# Characteristics of Authigenic Minerals around the Sulfate-Methane Transition Zone in the Methane-Rich Sediments of the Northern South China Sea: Inorganic Geochemical Evidence

**DOI:** 10.3390/ijerph16132299

**Published:** 2019-06-28

**Authors:** Daidai Wu, Tiantian Sun, Rui Xie, Mengdi Pan, Xuegang Chen, Ying Ye, Lihua Liu, Nengyou Wu

**Affiliations:** 1Key Laboratory of Gas Hydrate, Guangzhou Institute of Energy Conversion, Chinese Academy of sciences, Guangzhou 510640, China; 2Institution of South China Sea Ecology and Environmental Engineering, Chinese Academy of Science, Guangzhou 510301, China; 3Laboratory of Marine Mineral Resources, Pilot National Laboratory for Marine Sciences and Technology (Qingdao), Qingdao 266071, China; 4Ocean College, Zhejiang University, Zhoushan 316021, China; 5Key Laboratory of Gas Hydrate, Ministry of Naturaland Resources, Qingdao Institute of Marine Geology, Qingdao 266071, China

**Keywords:** authigenic minerals, anaerobic oxidation of methane, chromium reducible sulfur, methane seepage, South China Sea

## Abstract

Sediments at marine cold seep areas provide potential archives of past fluid flow, which allow insights into the evolution of past methane seepage activities. However, signals for anaerobic oxidation of methane (AOM) might be obscured in bulk sediments in cold-seep settings due to several factors, especially flood and turbidite deposition. Comprehensive inorganic data were gathered in this study to explore the availability of related records at cold seeps and to provide insights into the evolution of past methane seepage activities. Sediments collected from the site 973-4 in the Taixinan Basin on the northern slope of the South China Sea were characterized in terms of total carbon and sulfur, δ^13^C values of total organic carbon (δ^13^C_TIC_), δ^34^S values of chromium reducible sulfur (δ^34^S_CRS_), and foraminiferal oxygen and carbon isotopes. The results confirmed a strong correlation between formation of authigenic minerals and AOM. Moreover, the ^34^S enrichments and abundant chromium reducible sulfur (CRS) contents in the authigenic sulfides in the sulfate–methane transition zone (SMTZ) within 619–900 cm below seafloor (cmbsf) reflected past high methane fluxes supported by constant methane seepages. Lithological distribution and AMS (Accelerator Mass Spectra) ^14^C dating of planktonic foraminifera show that the turbidite (~35.14 ka) was related to a foraminifera-rich interval (Unit II: 440-619 cmbsf) and increased carbonate productivity during the last glacial maximum (LGM). Enrichment of Mo and U was observed accompanied by low contents of nutrient metals (Al, Ti, V, Ni, Fe, Mn, and Cu) in Unit II. The foraminifera-rich interval (Unit II) of cold seep sediments was probably linked to the phenomenon of inconsecutive sedimentary sequence due to the turbidites, which resulted in the lack of Fe, Mn, and Ba enrichment. There is no U enrichment but only Mo enrichment within Unit III, which might be related to H_2_S produced by AOM during the methane seepages. Based on the above results, it can be speculated that this area has experienced multiple-episodes of methane seep events. Further exploration of AOM should focus on the risks of rapid deposition, especially the impact of turbidity current on sediments.

## 1. Introduction

“Gas hydrate” (natural gas hydrate) is an ice-like crystalline compound composed of gas and water molecules which forms at low-temperature and high-pressure environments [1]. As their formation is dominated by methane gas in nature, gas hydrates are also known as methane hydrates. Gas hydrate occurs naturally in the sediments beneath the permafrost and the continental slope in water depths more than 300 m. Marine gas hydrate systems are important to economic society and the natural environment due to their viable energy potential as well as the geo-hazards they pose through large scale slope destabilization. Destabilization of gas hydrates will release methane, a potential greenhouse gas, which can affect the global climate [2,3,4]. Gas hydrate and its associated sediments have also become a focus of biogeochemical studies on the deep biosphere [5].

Marine sedimentary environment can be divided into several zones on the vertical cross-section according to the differences in concentrations of substances in marine sediments and the redox process in pore water. In fact, different geochemical zones usually overlap in spatial locations, and multiple oxidants and reaction products migrate, which also leads to dynamic changes in geochemical zones [6]. In addition, due to the content of oxidants such as NO_3_^−^, Mn_IV_, Fe_III_, and SO_4_^2−^ in pore water of marine sediments, the corresponding geochemical zone has different widths (centimeter or meter scale) [7]. In modern continental margins, sedimentary organic matter can be oxidized with dissolved oxygen and other oxidants. However, in areas with diffusive seepage, methane is mostly oxidized in the sediments at the expense of sulfate or metal oxidizers such as iron and manganese, by a series of biogeochemical processes [8,9]. In conclusion, when there is high methane flux in the sedimentary environment, the main biogeochemical reactions of concern are related to anaerobic oxidation of methane.

In cold seepage systems, large volumes of methane-rich fluids are transferred into seawater, where a series of important biogeochemical processes occur and sustain a broad diversity of ecosystems on the seafloor, relying on the energy provided by chemosynthetic microbes exploiting the oxidation of reduced chemical compounds [10,11]. The key biogeochemical process at seeps is the anaerobic oxidation of methane (AOM) coupled with sulfate reduction (SR) (1) [12].
(1)$${\mathrm{CH}}_{4}+{\mathrm{SO}}_{4}^{2-}\to {\;H}_{2}O+{\mathrm{HCO}}_{3}^{-}+{\mathrm{HS}}^{-}$$

This process produces dissolved inorganic carbon (DIC) and increases pore water alkalinity, thus favoring the precipitation of authigenic carbonates (2) [13].
(2)$${\mathrm{Ca}}^{2+}{+\mathrm{HCO}}_{3}^{-}\to {\mathrm{CaCO}}_{3}{+H}^{+}$$

The produced DIC during AOM is characterized by highly depleted ^13^C that is derived from methane [14,15]. At the same time, there is maximum accumulation of dissolved hydrogen sulfide in the sulfate–methane transition zone (SMTZ), where pore water sulfate and methane are depleted to near-zero concentrations [16,17]. The localized production of hydrogen sulfide commonly results in the precipitation of iron sulfide minerals (3) and (4) [18,19].
(3)$${\mathrm{Fe}}^{2+}{+\mathrm{HS}}^{-}\to {\mathrm{FeS}+H}^{+}$$
(4)$${\mathrm{FeS}+S}^{0}\to {\mathrm{FeS}}_{2}$$

Many studies have revealed the presence of ^34^S enriched authigenic sulfides in seepage areas. Moreover, such positive δ^34^S values were considered indicative of enhanced AOM–SR occurring at the SMTZ [20]. The carbon composition of upward migrating methane is mixed with other DIC sources, and the microbial preferences for ^32^SO_4_^2−^ often result in ^34^S-depleted sulfur in the pyrite [21]. Numerous studies have attempted to use geochemical proxies such as the content and isotopic anomalies of authigenic carbonates and pyrites to constrain methane seepage intensities and their variations [20,22,23,24].

Under different redox conditions, modern and ancient marine sediments are typically characterized by relative enrichment or depletion of various trace metals, such as the redox-sensitive metals Mo, U, and V and trace metals such as Cu, Fe, and Mn. Thus, they can be used as indicators of paleoredox conditions and paleoproductivity [25,26]. Although significant correlations exist between the concentrations of trace metals and sediment redox environments, it is unclear whether the relationship is predictable, particularly under non-steady state environments.

Here, a comprehensive inorganic geochemical data set is presented from the natural marine setting of Taixinan Basin using sediments collected during the cruise “Haiyang-6” in 2011. The non-steady state sedimentary environment along the continental margin of the northern slope of South China Sea is characterized by mass movements and attendant turbidity currents. Numerous periods of sea-level fall and lowstands have been reported [27,28]. The dynamics of fluid flows at seeps are characterized by changes in flow intensity and episodic seepages, which might be controlled by factors such as sea level variations, bottom water temperature fluctuations and the exhaustion of hydrocarbon sources that drive the dissociation of gas hydrates [29,30]. These particular depositional conditions have a significant impact on the biogeochemical processes of deeper marine sediments. In this study, we used only inorganic geochemistry to demonstrate past variations in methane fluxes. The findings of this study, would have significant implications for tracing past methane seepages, their intensities, and even possible occurrences of gas hydrate dissociation.

## 2. Materials and Methods

### 2.1. Geological Setting

The northern continental slope of the South China Sea extends from the southwestern end of Taiwan to the western end of the Xisha Trough in NE direction, with a total length of about 1350 km and a width of 143–342 km. The boundary between the northern continental slope and the deep-sea basin is 3400–3700 m, which is wide in the west and narrow in the east. The Pearl River and Hanjiang River in South China, Zeng Wenxi, and Gaoping River in Southwestern Taiwan all contribute significantly to the sediments in this area [31]. At the same time, because the South China Sea is located at 22 degrees north latitude, it is between the equator and the Tropic of Cancer. Thus, the monsoon climate is affected by the southwest monsoon in summer and northeast monsoon in winter. The driving force of the surface current in the South China Sea is influenced by the monsoon. In summer and winter, the Luzon Strait on the southern side of Taiwan Island ensures that there will be a sustained Kuroshio branch current entering and flowing to the southwest. Therefore, the Kuroshio also has a great impact on the sedimentation of the northern continental slope of the South China Sea [32]. In addition, there are some small asymmetric folds in the southeastern margin of the northern continental slope of the South China Sea. The development of these folds also has a significant influence on the sedimentary process of the northern continental slope of the South China Sea [33].

The northern continental slope area of the South China Sea is the junction of different tectonic units. Thus, it has the characteristics of huge topographic slopes and relatively developed faults in underlying strata. Many faults cut through newer sedimentary layers and extend to the vicinity of seabed sediments, opening up favorable channels for gas migration from the lower part to shallow formation, while fold structures can capture natural gas more easily, thus forming methane seepage development areas [34,35]. In May 2007 and June–September 2013, a variety of hydrate samples were drilled in Shenhu area of Baiyun Depression on the northern slope of the South China Sea and near the central uplift of the Southwestern Taiwanese Basin [36,37]. The drilling of these hydrate samples indicated that the northern part of the South China Sea was indeed a gas hydrate occurrence area. Moreover, the discovery of a large number of methanogenic carbonate nodules showed that methane escaped from the area [38]. The Taixinan Basin, also known as the Tainan Basin, is located to the east of Dongsha Islands in the Northeastern South China Sea. It is about 400 kilometers long and 150 kilometers wide with an area of more than 60,000 square kilometers. The complex submarine topography, widespread existence of folds, mud diapers and landslides, as well as fault-fold systems, all create a favorable tectonic environment for the development and migration of hydrocarbons and hydrates [39,40]. The Taixinan Basin contains thick layers of Late Cenozoic sediments through submarine canyons, up to 10 km thick, which accumulate in the form of turbid fluids, natural levees and sand-rich sediments such as submarine fans [41].

### 2.2. Materials and Methods

A 13.75 m long sediment column from site 973-4 was retrieved by the “Haiyang-6” cruise ship in 2011 from a water depth of 1666 m near the Jiulong Methane Reef in the Taixinan Basin, Northern South China Sea (SCS) (118°49.0818′ E, 21°54.3247′ N) (Figure 1). The piston core was then divided into 63 samples, every 20 cm below seafloor (cmbsf). A sample was selected every 20 cmbsf in the 0–1375 cmbsf interval, and a total of 63 samples were used for foraminifera community studies, with no samples at 558–560 cmbsf, 1038–1040 cmbsf, 1158–1196 cmbsf and 1239–1241 cmbsf. Each sample was freeze-dried, and some samples were powdered manually in an agate mortar for subsequent geochemical analysis.

Chromium reducible sulfur (mainly FeS, FeS2) found in samples was extracted using a modified method of [43]. This reduction method can release sulfur in all sulfide lattices, and the obtained sulfur content was almost the total content of chromium reducible sulfur (CRS) [6,43]. The reduced sulfur was converted to H_2_S gas, which was blown out by the carrier gas (nitrogen), and then, precipitated by passing through AgNO_3_-NH_3_H_2_O solution to obtain silver sulfide. The silver sulfide was filtered, dried and weighed. Then, the weight percentage of chromium reducible sulfide (CRS) in the deposit was calculated according to the weight of precipitated silver sulfide. Detailed experimental steps can be found in the report by Pan [42].

During the experiment, the quantitative addition of pyrite as a standard sample resulted in the recovery rate of 88–92%. The silver sulfide was then sent to the State Key Laboratory of Biogeology and Environmental Geology of China University of Geosciences (Wuhan, China) for the analysis of δ^34^S using an elemental analysis—isotope ratio mass spectrometer (DELTA V PLUS, Semerfly Technology (China) Co.; Ltd, Shanghai, China). The standard deviation of the measurement was less than 0.2‰ Vienna-defined Canyon Diablo Troilite (VCDT). All results are reported here in standard delta notation as per mil deviations from the Vienna-defined Canyon Diablo Troilite (VCDT). Measurement errors of ~0.2‰ (1σ) were calculated from replicate analyses of the IAEA international standards: IAEA S1 (−0.3‰), IAEA S2 (+22.7‰), and IAEA S3 (−32.3‰).

All sixty-three freeze-dried sediment samples were ground into uniform powder in agate mortar. Total organic carbon (TOC) was measured by ElementarTM Vario with an accuracy of 1% (5 mg/L). The powdered samples were determined by ElementarTM Vario EL cube elemental analyzer (Elementar Analysensysteme GmbH, Langenselbold, Germany). The average values of each sample were obtained twice. The analytical accuracy of total carbon (TC) and total sulfur (TS) was 0.1%. The total inorganic carbon (TIC) was mainly calcium carbonate. The calculation method of mass fraction was C(_CaCO3_) = (C_TC_ − C_TOC_) * 8.33 [44].

The mass fraction of foraminifera in sediments was calculated as follows: The mass fraction of foraminifera in sediments was estimated based on the average mass of foraminifera. Three samples of foraminifera from site 973-4 with different depths were selected: 973-4-83 sample from the upper part of the site, 973-4-223 sample from the middle part and 973-4-383 sample from the lower part. One hundred foraminifera of different sizes and species were selected from these different samples and weighed accurately. The average mass of foraminifera was calculated, and then the mass fraction distribution of foraminifera with depth was calculated using the content data of foraminifera.

The elemental analysis of all 63 freeze-dried sediment samples was performed using X-ray fluorescence (XRF) on a Thermo ARL ADVANTta IntelliPowerTM 2000 spectrometer (Thermo Fisher Scientific, Waltham, USA), which was available at the Zhejiang University of Technology, China. The relative standard deviation of the measurement was less than 5%. A total of 63 sediment samples were analyzed for trace elements by Agilent 7700e ICP-MS (Sample Solution Technology Co., Ltd., Wuhan, China) in the Analytical and Testing Center of Wuhan Shangpu Analysis Technology Co., Ltd. The treatment of sediment samples and the specific testing methods for major and trace elements have been described elsewhere [42]. To compare the respective authigenic enrichments of the trace metals in the sediment, enrichment factors (EF) were used, which are defined as XEF = [(X/Al) sample / (X/Al) PAAS], where X and Al represent the weight concentrations of elements X and Al, respectively. The samples were normalized using the Post Archean Australian Shale (PAAS) compositions [45]. In general, XEF N 3 represent a detectable authigenic enrichment, and XEF N 10 represent a moderate to strong enrichment [25].

## 3. Results

### 3.1. Core Description and Radiocarbon Dating

The core section was composed of four lithological parts (Figure 2). Unit I at the depth of 0–440 centimeters below sea floor (cmbsf) mainly contained celadon silty clay. Unit II at 440–619 cmbsf mainly consisted of gray clayey silt. In particular, massive foraminiferal sand was found in 479–494 cmbsf, and black iron sulfide stains were found in 494–510 cmbsf sediments. The middle part (530–603 cmbsf) was composed of gray-green clayey silt with visible foraminiferal shells. The low part at 619–900 cmbsf (Unit III) and 900–1365 cmbsf (Unit IV) were mainly composed of dense gray silty clay with different degrees of black hydrogen sulfide. It was noteworthy that the color and grain size of sediments at 603 cmbsf were clearly defined. Below 619 cmbsf, the core section emitted pungent gases such as hydrogen sulfide [46].

The AMS ^14^C dating results (Figure 2) of planktonic foraminifera at site 973-4 were collected according to Lin [47] and Pan [42]. The planktonic foraminifera (*N. dutertrei*) collected from 12 samples throughout the site yielded an inconsecutive sedimentary sequence, especially in Unit II. Unit I of celadon silty clay (0–440 cmbsf) increased in age with depth from 0.5–18.45 ka B.P, while the foraminifera-rich Unit II (440–619 cmbsf) was deposited between 34.09–39.65 ka B.P. Below the 619 cmbsf part, the core section recorded ages of around 19.00–25.94 ka B.P. The inconsecutive sedimentary sequence (Unit II) at this site may be related to turbidity current and other sedimentary events or the activities of underlying gas hydrate reservoirs.

### 3.2. Relevant Data of Carbonates in Bulk Sediments

The contents of CaO, TC and TIC from site 973-4 (Figure 3) exhibited similar geochemical behavior and were relatively stable throughout the core. Their values fluctuated at around 4 wt.%, 1.45 wt %, and 1.12 wt.%, except for obvious variations in Unit II, with the maximum values reaching 23.3 wt.%, 5.63 wt%, and 5.43 wt%, respectively. Total organic carbon (TOC) content ranged from 0.18 wt.% to 1.3 wt.% with a few appreciable changes throughout the core.

### 3.3. Chromium Reducible Sulfur and δ^34^S_CRS_

The chromium reducible sulfur (CRS) content at different depths measured at site 973-4 showed significant differences (Figure 4) (Appendix A
Table A1). The content of CRS in Unit I was relatively low, ranging from 0.072 to 0.139 wt.%, with an average value of 0.113 wt.%. The CRS content exhibited an almost linearly increasing trend with depth in Unit II, ranging from 0.123–1.385 wt.%, with an average of 0.376 wt.%. In Unit III, CRS content remained at a relatively high level with a maximum value of 1.211 wt%. However, in Unit IV, CRS content began to decline and remained relatively stable, ranging from 0.023–0.183 wt%. The profile of δ^34^S_CRS_ was similar to that of CRS content. The δ^34^S_CRS_ values ranged from −41.14‰ ~ 23.56‰. Above 619 cmbsf, δ^34^S_CRS_ values decreased from enriched to depleted isotopic ratios.

### 3.4. Concentration Profiles of Major and Trace Elements

The contents of selected major and trace elements in sediments of site 973-4 are presented in Figure 5. 

The contents of Al, Ti, Fe, Mn, and Ba in Unit II were lower compared to the other parts, and their average contents were 9.58%, 27.11%, 3.75%, 0.057%, and 433.33 ppm, respectively. The Mo and U contents showed obvious differences. The Mo contents ranged from 0.44 to 1.12 ppm with depth of the core, while the U contents ranged from 2.62 to 4.18 ppm. However, only Unit II exhibited moderate enrichment of Mo and U. The ratios of V/Sc showed no anomalies in the core except for Unit III, where the contents were slightly lower.

## 4. Discussion

### 4.1. Evidence of Anaerobic Oxidation of Methane in Sediments

The fluxes and rates of methane fluids released by the decomposition of deep gas hydrates vary in different areas of the sea and different geological periods. Based on this, the migration modes of methane fluids can be roughly divided into two types: diffusion type and seepage type. The former is usually in the form of slow release, where the methane fluid diffuses upward or around the sediment pores. The latter is in the form of intense seepage, where a large amount of methane fluid seeps upward along the sediment pores or structural fissures. The types and contents of authigenic minerals formed by different migration types of methane fluids are also different. The sulfate–methane transition zone (SMTZ) is usually located in the shallow surface sediments or even in the water body in sedimentary environments having intense seepage of methane. The strong anaerobic oxidation of methane (AOM; reaction (1)) produces HCO_3_^−^, which promotes the formation of authigenic carbonates with extremely negative ^13^C values (generally less than −25‰ VCDT) [50,51]. AOM–SR occurring in the SMTZ mainly results in the formation of authigenic iron sulfide minerals in the sedimentary environment where methane seeps slowly. However, AOM might limit the precipitation of authigenic carbonates whereas it is beneficial for the final conversion of intermediate products such as S^0^, SO_3_^2−^, and S_2_O_3_^2−^ into pyrite [52]. The resultant authigenic minerals are potential archives of past fluid flow, especially in combination with dating results. Thus, they can provide insights into the evolution of past methane seepages activities [11,29,30].

#### 4.1.1. CRS and δ^34^S_CRS_

The burial of authigenic sulfide in anoxic sediments is primarily controlled by the activity of sulfate reducing bacteria [53]. It should be noted that the typical burial ratio (weight) of organic carbon to sulfur for normal marine sediments falls within a relatively narrow range (2.0–3.6) [54]. By correlation analysis of TOC and TS contents in site 973-4, it can be seen that the characteristics of different sections were obviously different (Figure 4). In Unit I and Unit II, the CRS contents were generally at a low level (average value of 0.245 wt.%). The δ^34^S_CRS_ values (−40.26 ~ −14.68 ‰VCDT) were strongly depleted, which can also be seen in other continental margin sediments (Figure 4), which may be a result of the disproportionation of microbial sulfur occurring close to the sediment–water interface [20], additional sulfur cycling through the sulfate reduction zone [55], and/or slow sulfate reaction rates [56]. The similar trend ofδ^34^S_CRS_ values and δ^34^S_Pyrite_ values (Figure 4) suggest that iron–sulfide minerals represent the most important form of sulfur in sediment and that the TS content can be used to represent the trend of pyritization in sediment in site 973-4. A positive relationship between TOC and TS was observed. The average TOC/TS value was 2.634, close to 2.8, which belongs to normal marine sedimentary environment. This indicates that the CRS content in the sediments of Unit I and Unit II were mainly controlled by active organic matter through organoclastic sulfate reduction. However, the TOC/TS ratios of sediments within Unit II and below 619 cmbsf range did not follow this trend. They were lower than the typical average ratio (2–3.6), and in particular, the average TOC/TS ratio was only 0.71 in Unit IV. On the other hand, the contents of CRS and δ^34^S_CRS_ increased dramatically within Unit III while they remained at a relatively high level below Unit III. This indicates that the consumption of sulfate is increasing, or even relatively insufficient, which leads to excessive consumption of ^34^S in sulfate, resulting in a rapid increase of ^34^S in sulfides, usually close to the value of sulfate in seawater. This was a pattern similar to those found in the Nankai Trough and northern SCS, where the occurrence of AOM significantly increase CRS contents and δ^34^S_CRS_ values of the sediments [6,57]. Therefore, the observed high CRS content and δ^34^S_CRS_ values within depths of Unit III may indicate the intensification of AOM–SR and the current and/or past locations of the SMTZ. It has been well documented that the occurrence of the AOM at the SMTZ within Unit III could produce additional H_2_S, which diffused upward and thus decreased the TOC/TS ratio of sediments within Unit II [23]. A low CRS content was observed only at Unit IV, which is coincidence with the average low TOC/TS ratio in this Unit. However, Liu et al [49] demonstrated that the main biogeochemical processes in Unit IV is controlled by iron mediated anaerobic oxidation of methane (Fe–AOM), which further confirms that of methane rich fluids are well developed in this site.

#### 4.1.2. Authigenic Carbonates Precipitation

Numerous surveys and researches have indicated that authigenic carbonates originating from cold seeps are highly depleted in ^13^C, which serves as an important indicator of past methane seepage and the source of seep fluids [25,30,53,58]. Based on TC, TIC, CaO contents and δ^13^C_TIC_ values (Figure 3) in the four intervals (0–440, 440–619, 619–900, and 900–1360 cmbsf), the sediment core at site 973-4 reflects background sedimentation, and different prominent biogeochemical processes in different intervals. In fact, Unit III sediments revealed highly ^13^C-depleted (extremely negative values) authigenic carbonates, which indicated AOM–SR. On the other hand Unit IV had relatively ^13^C-depleted (−4.3‰) authigenic carbonates and was sulfate-poor due to iron–AOM, as demonstrated by Liu [49]. Nevertheless, the sulfate concentrations and δ^13^C_TIC_ values may allow recognition of organoclastic sulfate reduction (OSR) in Unit I where the minimum δ^13^C_TIC_ value was −2‰. The TC, CaO, and TIC profiles exhibited similar tendencies in the sediments at site 973-4, except for Unit II, which had relatively high contents. This also means that the high carbon content in Unit II range was due to the abnormal increase in content of CaCO_3_. This abnormal increase in content of CaCO_3_ within Unit II needs to be further investigated. 

Before analyzing the source of carbonate minerals in the site 973-4, the abnormal increase in carbonate content in the depth range of Unit II was analyzed (Figure 3). Some studies [42,46,48] have shown that foraminiferal assemblages in this cold seep site display high species abundance and diversity. A total of 9111 benthic foraminifera were obtained from 63 identified samples [48]. In addition, the samples also contained a large number of planktonic foraminifera and a small number of Ostracoda individuals. In the shallow depth of Unit II, foraminifera biomass increased significantly. Within the range of 458–618 cmbsf sampling depth, foraminifera increased sharply and reached the maximum at a depth of 459 cmbsf. The average abundance of foraminifera in this horizon reached 47 pieces/g, which was much higher than that of other Units. Therefore, it is speculated that the abnormal increase in CaCO_3_ in this area was not caused by the input of terrestrial materials, but probably by the increase in foraminiferal contents.

Benthic foraminifera inhabiting water–rock interfaces or sediments can effectively record changes in dissolved inorganic carbon in the surrounding environment [42,47,59]. According to the carbon and oxygen isotopes of benthic foraminifera (Figure 2) at site 973-4 reported by Zhang [48], the variation of CaCO_3_ contents with depth was compared. It was found that Unit II has recorded deposition since Marine Isotope Stage 2–3 (MIS2–3) period including the last glacial maximum (LGM). The carbon and oxygen isotope compositions of benthic foraminiferal shells decreased rapidly, similar to that of δ^13^C_TIC_. It can be inferred that Unit II was the result of various sedimentary processes. In addition to early diagenesis, reductive fluids brought about by methane seepage in the bottom had a great influence on the contents of foraminiferal shells and authigenic minerals (authigenic carbonate, CRS) and their isotopic compositions.

### 4.2. Source Origin of Methane-Rich Fluids: Constraint of the Stable C–O–S Isotope

Excluding the impact of methane seepage in Unit II, the sediments displayed a sharp decrease in δ^13^C_TIC_ from −0.6 to −3.8‰, accompanied by a dramatic increase in δ^34^S_CRS_ from −44.3 to 2.55‰. The carbon and oxygen isotopes of benthic foraminifera (e.g., *Uvigerina spp*.) and planktonic foraminifera (*P. obliquiloculata*) at site 973-4 decreased gradually at Unit II. Moreover, the carbon isotope values were both negative, which was not observed in other parts (Figure 2). Pan [60] analyzed the characteristics of foraminifera community in sediments from site 973-4. The results showed that the performance and diversity of benthic foraminifera in Unit II reached the maximum, and the community composition was completely different from that in other parts. This high density and abundance of foraminifera and carbonates also indicated that the stratum might be disturbed by turbidity current. Considering the fact that the overall δ^13^C_TIC_ value of sediments was much closer to biogenic end–member, it is suggested that the elemental C, O, and S in this interval likely originated from other sources than methane.

The lithological analysis of site 973-4 (Figure 2) shows that the sediments in Unit II were mainly clayey and silty, with large amounts of aggregated foraminiferal sand, shell debris, and carbonate blocks. Generally, the grain size of sediments increased compared with other intervals, which is the key characteristic of turbidity current. Several studies have found that the ^14^C age of sediments deposited from the cold seepage system may represent the maximum age of carbonate growth [59,60]. Nevertheless, the ^14^C dating in this study still provided a constraint for the carbonate growth. The planktonic foraminifera (*N. dutertrei*) throughout the site yielded an inconsecutive sedimentary sequence. It is therefore suggested that an abrupt increase in methane flux occurred between 14 ka and 19 ka, bringing sediments with an older age of 34.09–39.65 ka. The sampling interval of Unit II might have been affected by turbidity current [46]. During this period, there was a dramatic drop in the sea level all over the world, and the sea level was 120 m lower in the LGM than that at present [61,62]. The ESC sea level even suffered a drop of up to 150–160 m [63], and most of the continental shelf areas were exposed to the atmosphere. Some large rivers, including the Yangtze, are reported to have emptied directly into the western slope of OT [64,65]. The turbidites mainly developed during periods of sea level fall and lowstand in the northern and southern SCS [66]. Changes in pressure and temperature as well as the activation of fluid channels [29] likely resulted in extensive instability and decomposition of natural gas hydrates in the main continental shelf. Numerous studies have demonstrated that vigorous hydrocarbon fluid seepages occurred all over the continental shelf globally during this period, including the Black Sea [67], the Eastern Margin of Japan Sea [59], the Gulf of Mexico [29] and the South China Sea [68,69]. Due to the limitations of the AMS ^14^C dating method, it can be tentatively speculated that the gas hydrate dissociation in the study area was related to this global sea level falling event. Moreover, compared to the Holocene, the productivity of surface waters on the northern continental slope of the SCS was significantly higher during the last glacial maximum [70]. Since the last glacial maximum, the bulk carbon and sulfur isotopic data suggest that the SMTZ was relatively stable at roughly the same depth with a continuous but decreased flux of methane. Foraminiferal abundances can increase due to a sudden input of high quality organic matter, or enhanced reproductive activity and/or rapid growth [71]. This late transport in dynamic environments and the mixing of pore fluids likely affected the C–O–S isotope composition of foraminiferal communities and various authigenic minerals. Similar trends were observed in deep-sea basins in the western SCS and Dongsha area on the northern slope of the SCS, which are considered to be the result of turbidity currents [72].

### 4.3. Conditions for Mo Enrichments in the Methane Seep Environments

In this study, redox-sensitive elements Mo and U exhibited non-negligible anomalies in response to the intense methane seepages recognized in Unit II of the profiles (Figure 5), while the contents of these elements remained at a high level without abnormal changes in Unit IV. Alternatively, U_EF_ and V/Sc exhibited a range of responses to the intense methane seepages and showed detectable to moderate enrichments in the seep-impacted sediments in Unit II. U was strongly enriched in the seep-impacted sediments within Unit II. There were two Mo enrichments in Unite II and III. The upper one is at 479 cmbsf within Unit II where the Mo content of 0.94 ppm was observed, while the lower one is between 619 and 679 cmbsf within Unit III.

Previous studies suggest that the two key factors that may control redox-sensitive enrichments are the abundance of a metal in the terrigenous background sediment and the aqueous concentration of the metal in the overlying seawater [25]. However, the contents of terrigenous metals, such as Al and Ti, also influence the trace metal enrichment factors [26]. Due to the low Al and Ti contents, the high enrichments for Mo and U in Unit II with foraminifera-rich interval cannot be ascribed to the low Al content of sediments. Compared to the other trace elements Fe, Mn, and Ba had the lowest contents in the terrigenous background, which showed mainly suboxic conditions. Therefore, the Mo and U contents in this interval were unlikely to be influenced by the terrigenous background, and were instead influenced by suboxic conditions.

In modern continental margins, Mo enrichments commonly develop in TOC-rich surface sediments [73]. Under such environments, other redox sensitive elements (e.g., Ni, Cu, and Zn) that are closely associated with organic matter also show corresponding enrichments [73,74]. However, there is also a limited flux of trace elements to the sediment, or upward movement of the paleo–sulfate–methane transition zone (SMTZ). In this study, Mo and U enrichments occur in foraminifera-rich interval (Unit II) (Figure 6), and redox sensitive elements, such as Fe, Mn, Ba, and V are not enriched (Appendix A
Table A2). The foraminifera-rich interval (Unit II) of cold seep sediments was probably linked to the phenomenon of inconsecutive sedimentary sequence due to the turbidites, which resulted in the lack of Fe, Mn, and Ba enrichment. Anschutz et al. [75] reported that the redox conditions of sediments under turbidity current became anoxic only four months after turbidite deposition, which explained the development of reducing environment. Deep-sea turbidites, which typically non-steady state conditions and are widely found in the geologic record, are deemed to be an ideal ‘‘natural laboratory” to study the postdepositional mobility of redox-sensitive metals [3]. Therefore, the enrichment of U and Mo in foraminifera-rich interval may be related to turbidite deposition, which slowed oxygen diffusion into the sediment, allowing anoxic environments to occur in the sediments. 

However, it is worth noting that there is no U enrichment but only Mo enrichment within Unit III, which might be related to H_2_S produced by AOM during the methane seepages. In fact, AOM coupled with sulfate reduction can produce abundant H_2_S at cold seeps and generate sulfidic environments that favor Mo enrichments in sediments and result in high CRS content and δ^34^S_CRS_ values within depths of Unit III, as also proposed by recent studies [22,76,77,78]. 

### 4.4. Implications for the Dynamics of Past Methane Seepages

From the above analysis, it can be seen that site 973-4 recorded more methane seeps since 35 ka. Here, a conceptual model (Figure 6) is presented to simply describe the evolutionary stages of abundant authigenic minerals inducted by AOM. In the first stage (~35.14 ka), lithology, ^14^C dating and carbon and oxygen isotopes of foraminifera in the sedimentary column indicated that Unit II may have developed turbidity currents. It is therefore suggested that an abrupt increase in methane flux occurred between 14 ka and 19 ka, bringing sediments with an older age of 34.09–39.65 ka. Since the last glacial maximum, the bulk carbon and sulfur isotopic data suggest that the SMTZ was relatively stable at roughly the same depth with a continuous but decreased flux of methane. Foraminiferal abundances can increase due to a sudden input of high-quality organic matter, or enhanced reproductive activity and/or rapid growth, which provided possible conditions for the main biogeochemical process of AOM in sediments. The enrichment of U and Mo in foraminifera-rich interval may be related to turbidite deposition, which slowed oxygen diffusion into the sediment, allowing anoxic environments to occur in the sediments. These observations suggest that the U has possibly experienced remobilization as a consequence of downward penetration of the oxidation front from the oxygen-rich bottom currents. This pattern indicates that physical reworking (turbidity currents and bottom currents) of the seabed [3,79], which is common on continental margins, can significantly influence the behavior of U and other redox-sensitive and/or sulfide-associated metals. In the second stage (~19.0 ka), seeping of methane-rich fluids was also observed in Unit III, where most of the seeping methane was consumed by AOM–SR. Due to the high content of CRS and the enrichment of ^34^S, the flow of methane fluids was stable. In summary, the characteristics of methane seepages can be determined according to the composition and isotope characteristics of authigenic minerals in marine sediments. The analysis of sediment geochemistry should consider turbidites and other non-steady state factors.

## 5. Conclusions

Detailed data of authigenic minerals, inorganic elements and stable isotopes were reported for cold-seep sediments at the site 973-4 in the northern slope of the South China Sea, along with the associated grain size and AMS^14^C dating of planktonic foraminifera. This suggests that an abrupt increase in methane flux occurred between 14 ka and 19 ka, bringing sediments with an older age of 34.09–39.65 ka. The sampling interval of Unit II might have been affected by turbidity current. Therefore, the formation of the foraminifera-rich turbidites reported here might be closely related to the gas hydrate dissociation and increased carbonate productivity during the last glacial maximum, which provides a broad context for the settings and specific controls that facilitate AOM. The foraminifera-rich interval (Unit II) of cold seep sediments was probably linked to the phenomenon of inconsecutive sedimentary sequence due to the turbidites, which resulted in the lack of Fe, Mn, and Ba enrichment. The ^34^S_CRS_ enrichments and abundant CRS contents in the authigenic sulfides in SMTZ within Unit III (619–900 cmbsf) reflected the constant methane fluxes supported by constant fluid flow rate. The dynamics of fluid flows at seep were characterized by changes in flow intensity and episodic seepages, which might control the species, composition and formation depth of authigenic minerals.

## Figures and Tables

**Figure 1 ijerph-16-02299-f001:**
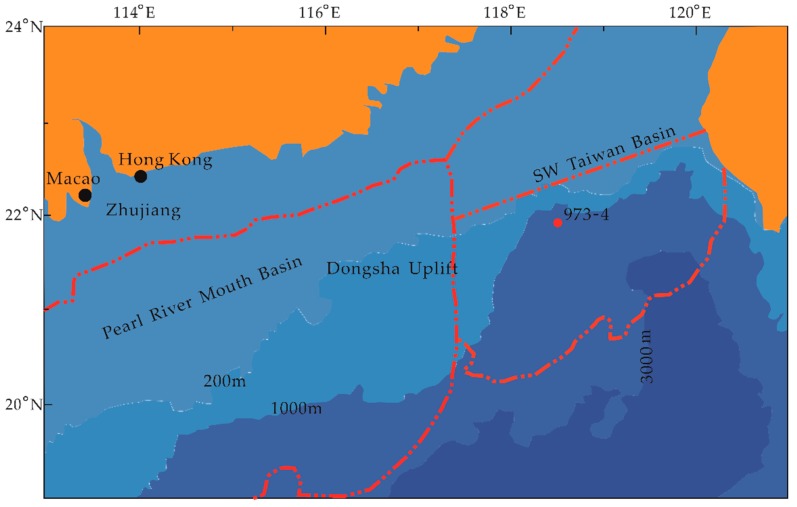
Map showing the locations of the study site 973-4 on the northern slope South China Sea (Modified from Pan et al. [42]).

**Figure 2 ijerph-16-02299-f002:**
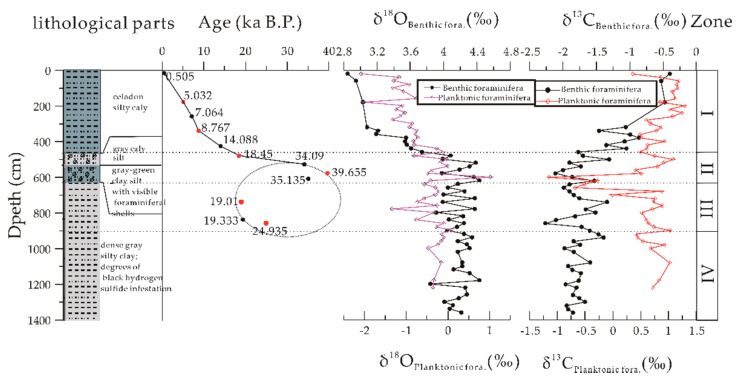
The lithological distribution [46] and AMS ^14^C age datum from site 973-4. Red dots represent the data of [42], while black dots represent the data of Lin [47]. Profiles of carbon and oxygen isotopic composition of benthic foraminifera (e.g., *Uvigerina spp*.) and planktonic foraminifera (*P. obliquiloculata*) at site 973-4 [48].

**Figure 3 ijerph-16-02299-f003:**
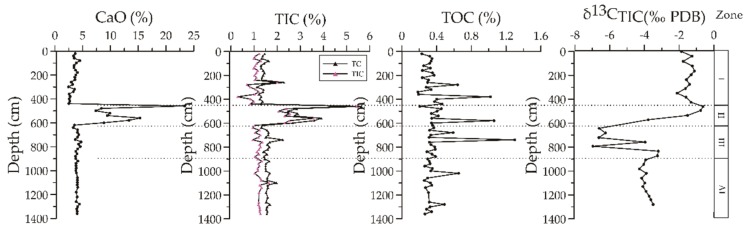
Depth profiles of Cao, total inorganic carbon (TIC), and total organic carbon (TOC) as well as carbon isotopic composition of total inorganic carbon. Carbon isotopic composition of TIC are adopted from Zhang et al. [49].

**Figure 4 ijerph-16-02299-f004:**
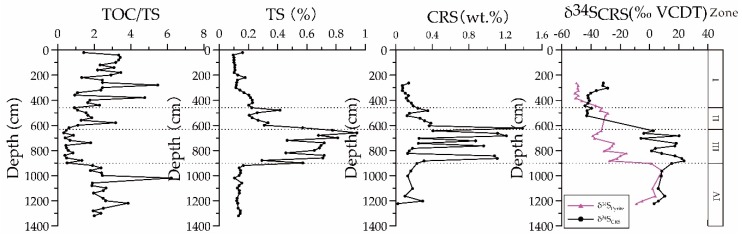
The contents of total sulfur (TS) and chromium reducible sulfur (CRS), and TOC/TS ratios, and sulfur isotopes of CRS (δ^34^S_CRS_) with depth in sediments of site 973-4. δ^34^S_Pyrite_ values of bulk sediments in site 973-4 were taken from Lin et al. [47].

**Figure 5 ijerph-16-02299-f005:**
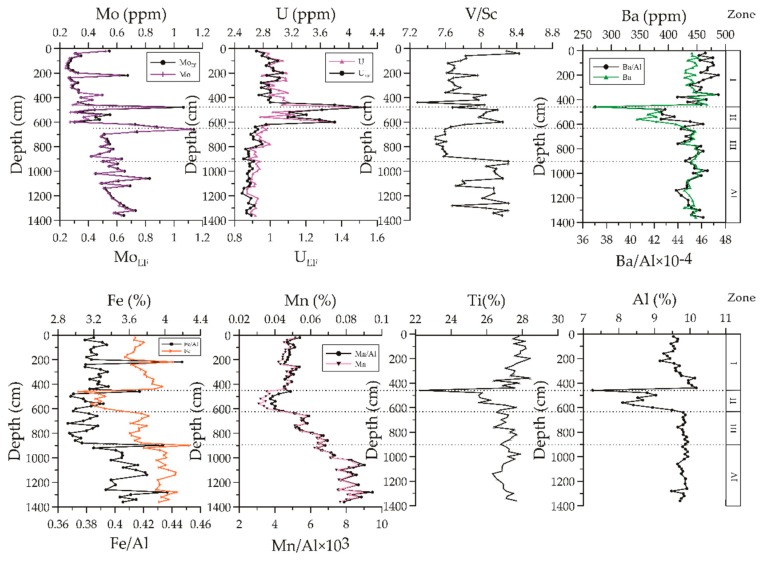
Depth profiles of Fe, Mn, Ti, Al, Mo, U, Ba, and ratios of Fe/Al, Mn/Al, Mo_EF_, U_EF_, Mn/Al, V/Sc, and Ba/Al in sediments of site 973-4.

**Figure 6 ijerph-16-02299-f006:**
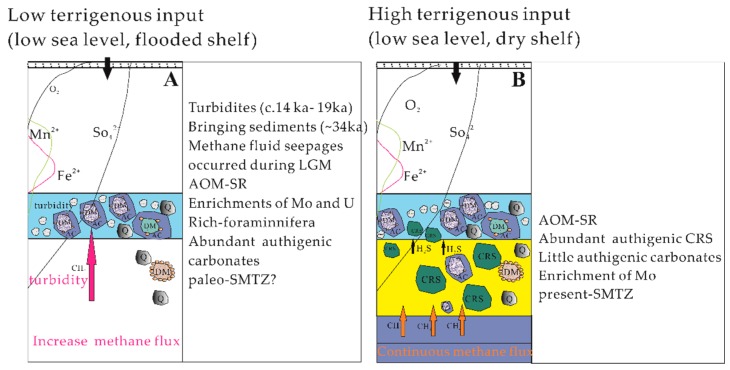
Schematic illustration of relevant depositional and diagenetic processes leading to methane seeping in continental slope setting affected by turbidity during (**A**) and after (**B**) the last glaciation. Q—quartz grain; DM—dark minerals; AC—authigenic minerals; CRS—chromium reduction sulfur; Fora.—foraminifera; AOM-SR: anaerobic oxidation of methane–sulfate reduction; SMTZ: sulfate methane transition zone; LGM: last glacial maximum.

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
