# Peer review of "Characteristics of Authigenic Minerals around the Sulfate-Methane Transition Zone in the Methane-Rich Sediments of the Northern South China Sea: Inorganic Geochemical Evidence"

_ijerph, 2019, doi:10.3390/ijerph16132299_

Round 1
Reviewer 1 Report
This manuscript by Wu and co-authors presents a comprehensive inorganic geochemical dataset in a core from the South China Sea. The authors discuss the characteristics of authigenic minerals around the sulfate-methane transition zone and the evolution of methane flux in the past. The bulk carbon and sulfur isotopic data suggest the occurrence of anaerobic methane oxidation coupled to sulfate reduction at a certain depth since the last glaciation. The overall quality of the paper makes it suitable for publication. However, several conclusions appear not to be supported by the data, which needs to be resolved before the manuscript can be accepted.
Scientific questions and issues:
1) Title. I found that “Cold-seep sediments” was not an appropriate term for this site. Cold seep, as it is named, is an area of the ocean floor where methane and other hydrocarbon-rich fluid seepage occurs. It has been shown that the SMTZ of cold seep sediments is usually shallower than 20 cm where methane concentrations are high in the sediment-water interface (SWI). The current SMTZ of this site is much deeper than those of cold seep sites. It is possible that methane flux suddenly increased during the Last Glacial Maximum. However, this does not imply a super shallow SMTZ near SMI. Moreover, regarding the dataset in this manuscript, there is no evidence suggesting that methane had reached the SWI. I understand the sediments were collected from cold seep area, but maybe not real seep, so I would suggest using methane-rich sediments instead of cold-seep sediments (e.g., Borowski et al., 1997, Marine Chemistry).
2) Line 251 and 252. (1) The occurrence of AOM-SR could indeed result in high sulfate reduction rate (SRR). It should be noted that, however, the sulfur isotope fractionation is correlated with cell-specific sulfate reduction rate (csSRR). A negative correlation exists between csSRR and sulfur isotope fractionation in pure cultures. However, most of these experiments were performed at much higher csSRR, than generally occur in marine sediments. (2) Sulfate availability also affects sulfur isotope fractionation. Theoretical work has suggested that the fractionation can remain large even under very low sulfate concentrations (Wing and Halevy, 2014, PNAS). Attenuation of sulfur isotope fractionation because of low sulfate may seldom be expressed under the low csSRR in marine sediments. (3) Taken together, the sulfur isotope fractionation should approach the thermodynamic equilibrium value between sulfate and sulfide in typical marine sediments (i.e. −70‰). The 34S-enriched pyrite was produced by porewater sulfate with extremely positive δ34S values via sulfate reduction. This is the result of a high sulfur isotope fractionation sustained even at the low sulfate concentrations found in the SMTZ. Please refer to a recent review paper, Jørgensen et al., 2019 in Frontiers in Microbiology.
3) Line 287 and 288. Debris of foraminifera is commonly found in marine sediments. It is most likely that those well-preserved large-size foraminifera only represent a minor fraction of the entire foraminifera pool deposited with burial. Most of foraminifera, either in form of debris or with small size, may escape the laboratory sieve, thus resulting in an underestimation of average mass of foraminifera. It is possible that, therefore, the abnormal increase in CaCO3 could be contributed by the elevated abundance of foraminifera in unit II. Other biogenic carbonate sources also should be taken into account, such as calcareous nannofossils. While the contribution of AOM-derived authigenic carbonate may be of minor importance in unit II, as reflected from the carbon isotopic composition of total inorganic carbon pool.
4) Line 311 and 312. The positive anomalies of Ca and Sr contents and negative Mg contents in Unit II are interpreted as the formation of aragonite. However, it is possible that the turbidites with increased biogenic carbonates in unit II are characterized by a similar geochemical property. Formation of aragonite is not necessary to interpret this abnormal layer.
5) Line 394. The authors should be careful when using the age of turbidites. Instead of c.35 ka, the turbidites in unit II were deposited between c.14 ka and c.19 ka (Figure 2). The age of turbidites does not imply the occurrence of unstable methane seeping activities around 35 ka. Instead, it might suggest that an abrupt increase in methane flux occurred between c.14 ka and c.19 ka, bringing sediments with an older age of c.35 ka. Since the Last Glacial Maximum, the bulk carbon and sulfur isotopic data suggest that the SMTZ was relatively stable at roughly the same depth with a continuous but decreased flux of methane (line 421, fluid flow rate is not constant on a long time-scale), as we take sedimentation into account.
6) Line 365 and 366. U/Al and V/Sc values are abnormally high in Unit II, which appears to display exact negative correlations with other trace elements (Fe, Mn, Ti, Al and Ba). I am in doubt whether the abnormal U/Al and V/Sc are authigenic. Regarding the lithology of unit II, it is possible that these U/Al and V/Sc are intrinsic to the original sediments that form the turbidites. This means, the enrichment of U and V might already exist under strongly reducing conditions in the c.35 ka sediments, which was brought to the location of this site by turbidity currents. I am not trying to deny the authors’ interpretation, while I found other explanation could also make sense.
7) Line 367 and 368. I found there were two Mo enrichments in Unite II and III. The upper one is at 479 cmbsf, while the lower one is between 619 and 679 cmbsf. Despite the limited data at 479 cmbsf, I found the board peak between 619 and 679 cmbsf to be particularly interesting. This lower peak is located below the turbidite layer, clearly suggesting that it is authigenic. It has been demonstrated that Mo enrichment is associated with the occurrence of high methane flux. However, this Mo enrichment alone does not imply the existence of cold seep. Other evidence such as authigenic carbonates with extremely negative carbon isotopic composition is required. Therefore, I would suggest that high methane flux had occurred in the upper unit III, possibly in unit II, during the Last Glacial Maximum. Moreover, please refer to Hu Y. et al., 2015 (Marine and Petroleum Geology), Chen F. et al., 2016 (Chemical Geology), Li N. et al., 2016 (Marine and Petroleum Geology) and Hu Y. et al., 2017 (Journal of Asian Earth Sciences).
8) Line 405 and 406. The influence of turbidites on sediment geochemical interpretation is not well-developed. It would be better to include some discussion about the impact of turbidites on sediment geochemical profiles. Please refer to Hensen et al., 2003 (GCA) and Hong et al., 2014 (Marine and Petroleum Geology).
9) Figure 7. The authors are mixing the original deposition that forms the turbidites at c.35 ka with the turbidites deposited between c.14 ka and c.19 ka. “Increased carbonate productivity”, “rich in foraminifera” and “limited flux of trace elements (Fe, Mn and Ba)” should belong to the original deposition at c.35 ka. Moreover, it is not clear whether there are abundant authigenic carbonates in unit II. Again, biogenic carbonates could be the main component of the increased TIC contents. Please clarify in this figure.
10). I have some key references of this manuscript. However, I found that some of them were not cited properly, including the published data. Please double check.
11) The figures are hard to read. Please replace with high resolution ones.
Detailed comments or suggestions are given below.
1) Line 115. The authors use “site 973-4” here but change to “973-4 site” in the following text. It would be better to use the former to make it consistent.
2) Line 131. Relative standard deviation of CRS concentrations analysis via extraction is not reported here. The quality of this dataset should be guaranteed.
3) Line 167 and 170. “Black hydrogen sulfide stains” are not appropriate since hydrogen sulfide has no color. “Black iron sulfide stains” would be better.
4) Figure 2. Change “carbon and oxygen isotopes of …” to “carbon and oxygen isotopic composition of …”. Please check throughout the manuscript.
5) Figure 4. The TOC profile is exactly the same as TC profile in the left. I found that it was wrong according to the dataset in the end. Please re-plot this figure.
6) Figure 5. The sulfur isotopic composition of pyrite is not mentioned in the text. Please clarify the difference between “pyrite” and CRS in the method and result sections.
7) Line 242 and 243. The negative sulfur isotopic composition of CRS does not suggest the occurrence of sulfur disproportionation. Assuming that the δ34S difference between pyrite and porewater sulfate represent sulfur isotope fractionation, such a large fractionation could be achieved by sulfate reduction alone without disproportionation, as suggested by Sim et al., 2011 (Science) and Leavitt et al., 2013 (PNAS).
8) Line 380. Change “pale” to “paleo”.
Author Response
Response to the reviewers’ reports and list of changes
Response to Reviewer 1 Comments
Dear reviewer,
We have thoroughly read your comments on our manuscript ijerph-522586 entitled "Characteristics of authigenic minerals around the sulfate-methane transition zone in the cold-seep sediments of the northern South China Sea: Inorganic geochemical evidence". We thank you very much for giving such a comprehensive feedback on our paper. We take these comments as very constructive to our work. According to your instructions, the corresponding issues were carefully addressed in the revised manuscript.

Reviewer 2 Report
The manuscript presented by Wu and co-workers is of interest to marine biogeochemists and environmental scientists. The methods used in the manuscript are appropriate for the purpose of their study and allowed a robust characterization of the sediment geochemistry. They discussed the main factors controlling the C-S systematics in seep impacted sediments in the South China Sea 973-4 site. The authors manage to get the message across, but the written English requires some improvement.
My main concerns are reported in the attached PDF and below:
The Authors report enrichment of Mo and U accompanied by low contents of nutrient metals (Al, Ti, V, Ni, Fe, Mn and Cu) in Unit II. Thus, it was hypothesized that the enrichments of Mo and U were due to the low Al and Ti contents of sediments, which was attributed to the influence of intense transient methane seepages as well as turbidites.” This is not clear to me. If I well understood, Mo and U enrichments are artifacts generated during normalization (caused by anomalously low Al), so how can you say it is an evidence for AOM?
It seems contradictory to me, please explain better your interpretation which is an important point of the paper.
The Authors should assess the effect of early diagenesis on the mass fraction of foraminifera tests
the Authors should explain in the method section the calculation of Mo and U enrichment factors. Did they use Tribovillard et al., 2012 Analysis of marine environmental conditions based on molybdenum–uranium covariation—Applications to Mesozoic paleoceanography ?
The Authors want to emphasize that organoclastic sulfate reduction and methane anaerobic oxidation are both sulfate-consuming processes but at cold seeps AOM is the dominant process consuming sulfate. Please try to make this concept easier to get. Now is quite confusing to me
I recommend substantial grammar revisions.
In the ABSTRACT Please consider rephrasing as follows: “Sediments collected from the 973-4 site in the Taixinan Basin on the northern slope of the South China Sea were characterized in terms of total carbon and sulfur, δ 13 C values of total organic carbon (δ 13 C TIC ), δ 34 S values of chromium reducible sulfur (δ 34 S CRS ) and foraminiferal oxygen and carbon isotopes were. The results confirmed a strong correlation between formation of authigenic minerals and AOM.”
INTRODUCTION Lines 11-12 please consider rephrasing as follows: “Gas hydrate” (natural gas hydrate) is an ice-like crystalline compound composed of gas and water molecules which forms at low- temperature and high-pressure environments.”
Lines 21-27: please state more clearly that it is a vertical redox zonation
Line 28: oxygen and oxidizers can only oxidize by definition, please remove “and reduced” from the sentence
29-32: I think that in this part
147-154 : The authors should assess the effect of early diagenesis on the mass fraction of foraminifera tests, e.g. SEM observations of tests may show if there is neomorfic calcite or some dissolution.
Other in the PDF

Author Response
Response to the reviewers’ reports and list of changes
Response to Reviewer 2 Comments
Dear reviewer,
We have thoroughly read your comments on our manuscript ijerph-522586 entitled "Characteristics of authigenic minerals around the sulfate-methane transition zone in the cold-seep sediments of the northern South China Sea: Inorganic geochemical evidence". We thank you very much for giving such a comprehensive feedback on our paper. We take these comments as very constructive to our work. According to your instructions, the corresponding issues were carefully addressed in the revised manuscript.
Point 1: The Authors report enrichment of Mo and U accompanied by low contents of nutrient metals (Al, Ti, V, Ni, Fe, Mn and Cu) in Unit II. Thus, it was hypothesized that the enrichments of Mo and U were due to the low Al and Ti contents of sediments, which was attributed to the influence of intense transient methane seepages as well as turbidites.” This is not clear to me. If I well understood, Mo and U enrichments are artifacts generated during normalization (caused by anomalously low Al), so how can you say it is an evidence for AOM? It seems contradictory to me, please explain better your interpretation which is an important point of the paper.
Response 1: Thank you for this valuable suggestion. The SCS has witnessed dramatic changes in depositional environments since the late Quaternary glacial cycles. These changes directly and indirectly altered the geochemical characteristics of elements in the sediments and authigenic minerals (authigenic sulfide) contained in them. We should pay attention to the influence of the degree of depositional environments restriction, diagenesis and postdepositional reoxygenation on the concentrations of these trace elements such as Mo and U etc. The conclusion that Mo and U enrichments occur in foraminifera-rich interval (Unit II) (Fig. 5), and redox sensitive elements, such as Fe, Mn, Ba and V are not enriched may be related to turbidite deposition, which slowed oxygen diffusion into the sediment, allowing anoxic environments to occur in the sediments. However, there is no U enrichment but only Mo enrichment within Unit III, which might be related to H2S produced by AOM during the methane seepages. We have rewritten the description of the sentence, and they are marked in red (Line 409-427).
Point 2: The authors should assess the effect of early diagenesis on the mass fraction of foraminifera tests.
Response 2: Thank you so much for this insightful comments. Although the foraminifera abundance are relatively higher in the Unit II than that in the other units. Debris of foraminifera is commonly found in marine sediments. It is most likely that those well-preserved large-size foraminifera only represent a minor fraction of the entire foraminifera pool deposited with burial. Most of foraminifera, either in form of debris or with small size, may escape the laboratory sieve, thus resulting in an underestimation of average mass of foraminifera. Our samples have been affected by early diagenesis. There is no doubt that more precise and detailed work should be done in future to reveal more direct and profound evidence for authigenic carbonates and methane activities.
Point 3: The authors should explain in the method section the calculation of Mo and U enrichment factors. Did they use Tribovillard et al. (2012). Analysis of marine environmental conditions based on molybdenum–uranium covariation—Applications to Mesozoic paleoceanography?
Response 3: Thank you so much for the comment. We have checked some papers on the definition of element Al in the ancient marine environment, especially those written by Porf. Calvert and confirmed that it is usually used as a normalization parameter representing the background value of terrestrial source. So we deducted terrigenous components. To compare the respective enrichment of Mo and U in the sediments, enrichment factors (EF) were used, which are defined as XEF = [(X/Al)sample/(X/Al)PAAS], where X and Al represent the weight concentrations of elements X and Al, respectively. The samples were normalized using the Post Archean Australian Shale (PAAS) composition (Taylor and McLennan, 1985). They are all based on Tribovillard et al (2012). So we have added the method of calculations in the method session. They are marked in red (Line 201-206).
Point 4: The Authors want to emphasize that organoclastic sulfate reduction and methane anaerobic oxidation are both sulfate-consuming processes but at cold seeps AOM is the dominant process consuming sulfate. Please try to make this concept easier to get. Now is quite confusing to me.
Response 4: Thanks for the advice and sorry for the carelessness. During early sediment diagenesis, organiclastic sulfate reduction (OSR) occurs, using organic matter and seawater sulfate as substrate and also yielding hydrogen sulphide. In areas of diffusive seepage, however, methane is mostly oxidized in the sediments at the expense of sulfate by a biogeochemical process known as sulfate-driven anaerobic oxidation of methane (AOM-SR). We have revised them.
Point 5: I recommend substantial grammar revisions.
Response 5: Thank you for your suggestion. We have sent the manuscript to native speakers from MogoEdit (a company that offers language editing services) for professional language editing. We really hope that this time my written English is improved.
Point 6: In the ABSTRACT Please consider rephrasing as follows: “Sediments collected from the 973-4 site in the Taixinan Basin on the northern slope of the South China Sea were characterized in terms of total carbon and sulfur, δ13C values of total organic carbon (δ13CTIC), δ34S values of chromium reducible sulfur (δ34SCRS) and foraminiferal oxygen and carbon isotopes were. The results confirmed a strong correlation between formation of authigenic minerals and AOM.”
Response 6: Thank you for your suggestion. We have revised this sentence in the abstract. They are marked in red.
Point 7: 1) INTRODUCTION Lines 11-12 please consider rephrasing as follows: “Gas hydrate” (natural gas hydrate) is an ice-like crystalline compound composed of gas and water molecules which forms at low- temperature and high-pressure environments.”
2) Line 28: oxygen and oxidizers can only oxidize by definition, please remove “and reduced” from the sentence.
3) The authors should assess the effect of early diagenesis on the mass fraction of foraminifera tests, e.g. SEM observations of tests may show if there is neomorfic calcite or some dissolution.
Response 7: Thank you for your comments and sorry for the misuse. We have checked the whole manuscript and made some modifications according to your suggestions. As for the mass fraction of foraminifera tests, we just cited by Zhang [48]. For more information, please refer to Zhang et al [48].
Point 8: Lines 21-27: please state more clearly that it is a vertical redox zonation
Response 8: Thank you for your comments. We have revised this sentence as “Marine sedimentary environment can be divided into several zones on the vertical cross-section according to the differences in concentrations of substances in marine sediments and the redox process in pore water”.

Reviewer 3 Report
Dear Authors
the paper is well written, the results are clear and interested. I only suggest to improve the discussion, figure quality and bibliography.
Author Response
Response to the reviewers’ reports and list of changes
Response to Reviewer 3 Comments
Dear reviewer,
We have thoroughly read your comments on our manuscript ijerph-522586 entitled "Characteristics of authigenic minerals around the sulfate-methane transition zone in the cold-seep sediments of the northern South China Sea: Inorganic geochemical evidence". We thank you very much for giving such a comprehensive feedback on our paper. We take these comments as very constructive to our work. According to your instructions, we have revised some of the discussion part. And all the references and figures are now rechecked and some mistakes have been modified.
With kind regards
XXX
